# Learning Normal Patterns in Musical Loops: Unsupervised Anomaly Detection for Variable-Length Audio Inputs

Shayan Dadman[*1], Bernt Arild Bremdal[1], Børre Bang[1], and Rune Dalmo[1]

[1]Department of Computer Science, UiT, The Arctic University of Tromsø, Lodve Langesgate 2, 8514 Narvik, Norway
 {shayan.dadman, bernt.a.bremdal, borre.bang, rune.dalmo}@uit.no

## Abstract

We propose an unsupervised framework for detecting audio patterns in musical loops using deep feature extraction and anomaly detection. Unlike prior methods limited by fixed input lengths, handcrafted features, or domain constraints, our approach combines a pre-trained Hierarchical Token-semantic Audio Transformer (HTS-AT) and Feature Fusion Mechanism (FFM) to generate representations from variable-length audio. These embeddings are analyzed by Deep Support Vector Data Description (Deep SVDD), which models normative patterns in a compact latent space. Experiments on bass and guitar datasets show our Deep SVDD models—especially with residual autoencoders—outperform baselines like Isolation Forest and PCA, achieving better anomaly separation. Our work provides a flexible, unsupervised method for effective pattern discovery in diverse audio samples.

## 1 Introduction

Musical loops are essential in modern music production, particularly in genres such as hip-hop and Electronic music [1, 2]. These repeatable audio segments provide rhythmic, melodic, or harmonic foundations [3]. Producers and DJs often search libraries or sample existing tracks for loops that match their artistic goals [4, 5]. Effective pattern analysis within loops is crucial for selecting compatible elements [6, 7] and identifying variations or inconsistencies in large or generated music collections.

AI's growing role in music requires interactive tools that emphasize user control. As highlighted in broader discussions on AI-driven music generation systems (MGS) [8, 9], there is a demand for technologies that move beyond 'black-box' paradigms [10]. Creators seek systems customizable to their workflows, effective with private data, and transparent in operation Civit et al. [11]. Our unsupervised pattern detection approach addresses these demands. We hypothesize unsupervised methods will adapt to user collections and styles without pre-labeled datasets (Section 7.1).

In Music Information Retrieval (MIR) [12, 13], loop detection constitutes a specialized audio pattern recognition task [14]. This encompasses subtasks like classification and tagging [15–17], as well as extracting features to identify and categorize signals [18]. Leveraging deep learning (DL) advancements, particularly pre-trained encoders that generalize across datasets [19], supports our hypothesis that such models can underpin loop analysis by capturing temporal and hierarchical details. Despite progress in DL for audio pattern recognition [14, 19], loop analysis methods often face limitations.

Traditional signal processing and structural analysis techniques [20–23] can struggle with the complexity of real-world music and often require heuristic adjustments. Neural networks may require fixed-length inputs [7, 24], restrict variable-duration loops, or need iterative feedback or labeled data [25], which limits scalability. These challenges define three core design criteria: variable-length support, unsupervised learning, and structural flexibility. Solutions should enable variable-length analysis, accommodate diverse structures, and work unsupervised without domain constraints.

To address these challenges, this paper poses three research questions: (1) How can unsupervised anomaly detection frameworks support adaptable, variable-length audio loop pattern analysis for user-specific music collections? (2) To what extent can such frameworks learn 'normative' patterns and identify meaningful deviations? (3) How do architectural choices influence performance in unsupervised audio pattern detection?

To address this, we propose framing loop pattern detection as an unsupervised anomaly detection problem. The insight is that normative patterns in a dataset or style can be learned from unlabeled data, enabling personalized adaptation without labeled sets. Anomalies—deviations from norms—offer insights into variations, errors for quality control, or stylistic elements fostering creative discovery (elaborated in Section 7). Our method integrates a pre-trained Hierarchical Token-semantic Audio Transformer (HTS-AT) [26], selected for capturing both local and global temporal dependencies, with a Feature Fusion Mechanism (FFM) [19] to produce a fixed-dimensional embedding for variable

---

*Corresponding Author.

Proceedings of the 7th Northern Lights Deep Learning Conference (NLDL), PMLR 307, 2026.

inputs. A Deep Support Vector Data Description (Deep SVDD) network [27] trains on these, learning a hypersphere enclosing normal instances.

In response to RQs, this work makes the following contributions:

- We introduce and implement an unsupervised framework for audio loop pattern detection, combining HTS-AT with FFM for flexible feature extraction, and Deep SVDD for anomaly-based identification directly from the learned audio embeddings. This approach offers a pathway to data-driven insights from user-specific collections, aligning with the need for more adaptable AI tools.

- We demonstrate the system's capability to process variable-length audio inputs through the FFM, thereby overcoming a limitation of many prior fixed-length approaches and enhancing practical applicability.

- We provide an empirical evaluation on curated datasets of bass and guitar loops, showcasing the model's proficiency in learning representations of normative patterns and effectively identifying meaningful deviations.

- We conduct a comparative analysis of different Deep SVDD encoder architectures (a standard Autoencoder versus an Autoencoder with Residual Connections), offering insights into architectural choices that benefit the modeling of diverse and larger audio data.

- We benchmark our proposed method against standard unsupervised anomaly detection techniques (Isolation Forest (IF) and PCA-based reconstruction error), demonstrating the enhanced representational power and discriminative ability of the Deep SVDD approach when operating on HTS-AT embeddings.

We position this work as a foundational proof-of-concept, demonstrating the framework's viability on string instruments (bass and guitar) to pave the way for future application to diverse musical timbres.

The remainder of this paper is structured as follows: Section 2 reviews relevant prior work. Section 3 details the proposed architecture and its components. Section 4 describes the datasets, preprocessing steps, and training procedures. Section 5 outlines the evaluation and baseline methods. Section 6 presents the experimental results. Section 7 discusses the findings and their implications, followed by Sections 8 and 9, which addresses limitations and outlines future research directions. Finally, Section 10 concludes this paper.

## 2 Related Works

### 2.1 Audio Pattern Recognition

Recent deep learning advances have significantly improved audio pattern recognition. Pretrained Audio Neural Networks (PANNs) [14], trained on large datasets such as AudioSet [16], provide robust features for tasks like audio tagging, scene classification, and event detection [28]. CNNs, ResNets [29], and MobileNets [30] are commonly used, typically processing Mel-spectrograms [14]. Although 1D CNNs on waveforms (e.g., DaiNet, LeeNet) have been explored, spectrogram-based models remain superior due to their ability to capture frequency patterns [14]. Wavegram-Logmel-CNN [14] integrates learned and hand-crafted time-frequency features, achieving state-of-the-art AudioSet tagging. Methods like FFM address variable-length inputs, showing promise for loop detection [19]. Newer models such as HTS-AT [19] further enhance sequential audio data modeling, outperforming earlier PANNs.

### 2.2 Loop Selection and Extraction

Early research on music loop identification relied on signal processing and rule-based methods, using handcrafted features (e.g., chroma, MFCCs) and heuristics to detect repeating patterns and estimate similarity [7, 31–36]. While approaches such as Non-negative Tensor Factorization (NTF) [36] and psychoacoustic modeling [33, 34] were explored, these techniques required extensive tuning and struggled with the complexity of real-world music [7]. Recent advances address these limitations with neural network-based solutions.

Building on the shift toward neural network approaches, Chen et al. [7] proposed NN models for estimating compatibility in large libraries. Their approach included a CNN on combined time-frequency representations of loop pairs and a Siamese Neural Network (SNN) comparing separate embeddings. The models were trained on Hip-Hop pairs from the Free Music Archive (FMA) dataset [37]. Loops were time-stretched to 2 seconds and converted to log mel-spectrograms. These models outperformed AutoMashUpper in subjective tests [7]. However, their use of fixed-length inputs limits handling of variable durations.

In a related line of inquiry, Jakubik [25] developed an active learning system for retrieving interesting loops and samples in electronic tracks. The system refined results via user interaction from an example. It compared MFCCs with unsupervised features from autoencoders and Bootstrap Your Own Latent (BYOL) contrastive learning [38], using `sampleswap`[1] data. Feature learning enhanced recall over MFCCs, especially on representative data;

---
[1] https://www.sampleswap.org/

however, performance varied by genre (e.g., Dubstep challenges) and relied on user feedback. This limited automation. Distinct samples were harder to find than repeating loops [25].

Shifting focus from audio features, Han et al. [24] addressed symbolic MIDI loop generation for 8-bar bass and drum loops via a two-stage process. First, a Vector Quantized Variational Autoencoder (VQ-VAE) compressed input into latent codes. Then, an autoregressive generator produced new material. A cross-domain loop detector, trained on 1,000 `looperman`[2] audio loops using One-Class Deep SVDD [27], identified domain-invariant patterns in bar-to-bar correlation matrices. The same detector was used to extract MIDI loops from the Lakh MIDI Dataset[3]. Limitations include fixed 8-bar windows restricting flexibility. Focusing on bar-to-bar correlation may miss higher-level audio nuances, similar to [7].

# 3 Methodology

This section presents our methodology for identifying repetitive patterns in audio loops by framing the task as anomaly detection. We learn a representation of 'normal' loop structures, enabling detection of deviations as 'anomalies'. As shown in Figure 1, our system consists of two stages: an Audio Encoder and a Deep SVDD module. The Audio Encoder generates embeddings from input audio using FFM and HTS-AT models, which are then processed by the Deep SVDD. A detailed architecture illustration is available in Appendix A.

The Audio Encoder uses a dual-path strategy based on input duration to handle variable lengths and capture both local and global details. Inputs ≤ 10 seconds are repeated and padded for consistency; longer inputs are split into three 10-second segments for local analysis and a downsampled global segment. All audio is converted to Mel-spectrograms, processed by initial Conv2D layers, and, for longer inputs, local features are merged and fused with global features via Attention Feature Fusion (AFF; see Section 3.1). The resulting features, either from padded short inputs or fused long inputs, are passed to the HTS-AT model (Section 3.2). The Audio Encoder's output forms the embedding for the Deep SVDD module.

The Audio Encoder's embeddings are processed by the Deep SVDD module (Section 3.3), which consists of five encoder layers that map them into a lower-dimensional latent space. In this space, a hypersphere is learned to enclose 'normal' audio patterns, with anomalies lying outside. The embedding's distance from the center serves as a score for

[2] https://www.looperman.com/
[3] https://colinraffel.com/projects/lmd/

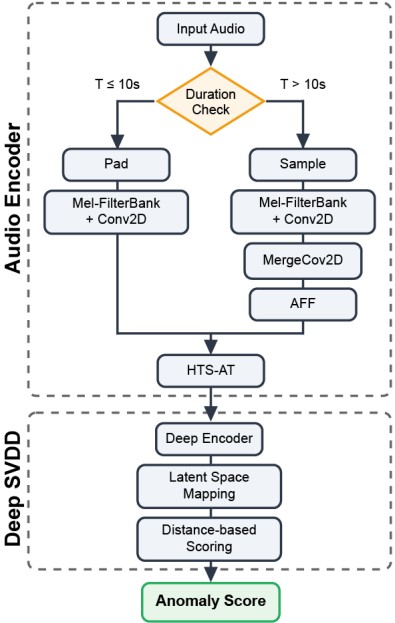

**Figure 1.** Overview of the loop detection pipeline. Input audio undergoes conditional preprocessing based on duration, followed by hierarchical feature extraction through Mel-FilterBank and Conv2D layers, feature fusion with pre-trained HTS-AT, encoding through 5 layers, latent space mapping, and distance-based scoring to produce the final anomaly score. See Appendix A for the detailed architecture diagram.

anomaly detection. Implementation details of each component follow in the next subsections.

## 3.1 Feature Fusion Mechanism

To address variable audio input lengths—a limitation in prior work (Section 2)—we implement the FFM, following Wu et al. This mechanism manages differing audio clip lengths in the Audio Encoder by ensuring consistent processing dimensions and integrating multi-scale information. Clips of $d$-seconds or less are repeated and padded to $d$-seconds; longer clips use a dual representation:

1. Global: Downsample the whole clip to $d$-seconds.

2. Local: Three $d$-second segments are randomly sliced from the beginning (first 1/3), middle (second 1/3), and end (final 1/3) of the clip.

Conv2D layers extract features from Mel-spectrograms. For long clips, features from the local segments are further consolidated via an additional Conv2D layer. Long-clip fusion uses the Attention Feature Fusion (AFF) module from Dai et al. Let $F_{local}$ denote the consolidated local features and $F_{global}$ the global features. The AFF module fuses these inputs by computing a dynamic, content-aware

weighted average. The final fused feature map, $Z$, is calculated as:

$$Z = M\big(F_{local} + F_{global}\big) \otimes F_{local} \\ + \big(1 - M\big(F_{local} + F_{global}\big)\big) \otimes F_{global} \quad (1)$$

where $\otimes$ represents element-wise multiplication. The attention map $M(\cdot)$ is generated by a Multi-Scale Channel Attention Module (MS-CAM) [39], which combines both local and global channel contexts to adaptively determine the fusion weights. This allows the model to prioritize the most relevant information from the dual representations. The fused map is then input to the HTS-AT module.

## 3.2 Hierarchical Token-semantic Audio Transformer

We employ HTS-AT [26] as our audio feature extractor due to its ability to efficiently model hierarchical audio structures. Its design integrates windowed attention, limiting self-attention to local $M \times M$ regions for computational efficiency, and patch-merging, which reduces sequence length in deeper layers. For an input of $ft$ patch tokens with latent dimension $D$, windowed attention achieves linear complexity in sequence length:

$$\Omega(\text{WA}) = \mathcal{O}(ftD^2 + M^2 ftD) \quad (2)$$

This approach avoids the quadratic complexity of global attention, which is a bottleneck for long sequences:

$$\Omega(\text{GA}) = \mathcal{O}(ftD^2 + ft^2 D) \quad (3)$$

Patch-merging follows groups of transformer blocks, merging adjacent patches (e.g., down-sampling a $\frac{T}{P} \times \frac{F}{P}$ token map with dimension $D$ to $\frac{T}{2P} \times \frac{F}{2P}$ and projecting to $2D$), which further reduces sequence length and computation.

Instead of using the HTS-AT classification head, we use the model as a feature extractor, aggregating output tokens from the transformer blocks into a single audio embedding vector. We employ pre-trained HTS-AT weights from the LAION-CLAP model (HuggingFace), keeping them frozen during Deep SVDD training (see Chen et al. [26]).

## 3.3 Deep Support Vector Data Description

The final stage of our system uses Deep SVDD [27], an unsupervised anomaly detection method based on classical SVDD [40]. We frame audio loop pattern analysis as anomaly detection: common or structurally coherent loop patterns are 'normal,' while novel or divergent ones are 'anomalies.'

Deep SVDD is well-suited for our task because its unsupervised approach enables direct learning from large, unlabeled audio loop collections. This supports our aim for adaptable systems without manual annotation. Deep SVDD also learns a compact, data-driven boundary of normality by mapping audio embeddings into a minimal hypersphere via neural network encoder layers (see Figure A.1). This approach lets the network capture the shared characteristics of 'normal' loop data and adapt to diverse musical characteristics, as demonstrated in related work [24].

The Deep SVDD module receives audio embeddings $\mathbf{z}$ from the encoder and maps them via a network $\phi(\cdot; W)$ to a latent space. The objective is to minimize the hypersphere volume (center $c$, radius $R$), enclosing most 'normal' patterns. This can be formulated as:

$$\min_W \frac{1}{N} \sum_{i=1}^{N} \|\phi(z_i; W) - c\|^2 + \frac{\lambda}{2} \sum_{l=1}^{L} \|W_l\|_F^2 \quad (4)$$

where the first term penalizes distances from the center $c$ for $N$ normal training samples, and the second term is a network weight decay regularizer (with $L$ layers and Frobenius norm $|\cdot|_F$). The center $c$ is often fixed as the mean of initial network outputs for the training data or can be learned.

During training on representative 'normal' loop patterns, the network $\phi$ is optimized to learn the common factors of variation, effectively pulling their latent representations towards the hypersphere's center $c$. Consequently, loop patterns that deviate significantly from these learned commonalities will be mapped further from $c$ in the latent space. The anomaly score $S(\mathbf{x})$ for a given input loop $\mathbf{x}$ (which yields embedding $\mathbf{z}$) is then its squared Euclidean distance to the center $c$:

$$S(x) = \|\phi(\mathbf{x}) - c\|^2 \quad (5)$$

A lower score indicates that the loop's characteristics closely resemble the 'normal' patterns learned during training, while a higher score signifies a deviation, marking it as 'anomalous' or distinct. This score is the final output of the Deep SVDD module, providing a quantifiable measure to identify potentially interesting, unusual, or structurally divergent audio loops within a collection.

# 4 Experimental Setup

## 4.1 Data

We curated a dataset of 6110 royalty-free guitar and bass WAV samples (2.34 hours bass, 5.92 hours guitar) from MusicRadar [41]. Due to redistribution restrictions, the dataset cannot be shared. Metadata

(genre, key, BPM) was extracted from file and folder names; BPM was refined using `deeprhythm` [42] when differing by over 10. Durations were calculated with `librosa`[4]. Most samples lack reliable genre/key labels due to metadata inconsistencies, but the dataset's diversity enables evaluation of the model's ability to distinguish typical from anomalous patterns. Appendix B details tempo, duration, and distribution statistics. We split the data into bass-only and guitar-only subsets with an 80/20 training/validation split for hyperparameter tuning and early stopping. No test set was reserved, as the focus was unsupervised learning. Instruments were analyzed separately to account for their distinct characteristics and to evaluate the impact of feature sets on Deep SVDD training.

## 4.2 Data Preprocessing

Audio samples were resampled to 48 kHz and converted to Mel-spectrograms using STFT (window size 1024, hop size 480, 64 Mel filter banks), following [19]. This yields $(T = 1024, F = 64)$ spectral representations for 10-second segments, as input to the Audio Encoder described in Section 3. Variable-length inputs were accommodated by the dual-path strategy (Section 3.1). A pre-trained, frozen HTS-AT encoder (Section 3.2) was used in inference mode, producing $(1, 1024)$ embeddings pre-computed for all samples and fed directly to Deep SVDD (Section 3.3).

## 4.3 Hyperparameters and Training Details

We implemented Deep SVDD using two neural network architectures: a standard autoencoder (AE) and an autoencoder with residual connections (AEwRES). Both architectures comprise multiple fully connected layers with ELU activations [43]. We employed dropout regularization and batch normalization. Training used two phases. In AE pre-training, the autoencoder learned a compressed data representation by training on pre-computed embeddings with a Mean Squared Error (MSE) reconstruction loss. For Deep SVDD fine-tuning, we discarded the decoder and fine-tuned the encoder using the Deep SVDD objective [27]. This minimized the volume of the hypersphere enclosing normal data embeddings. Both phases utilized the AdamW optimizer [44] with an initial learning rate of $1 \times 10^{-3}$ and weight decay of $1 \times 10^{-5}$. We paired this with a Cosine Annealing scheduler [45] and a minimum learning rate of $5 \times 10^{-6}$. Models trained for 1000 epochs with a batch size of 32. We applied early stopping with a patience of 20 by monitoring the validation

loss. The implementation utilized `PyTorch`, and experiments were tracked using `Weights & Biases` [5].

# 5 Evaluation

Evaluating anomaly detection without ground-truth labels necessitates focusing on representation quality and model behavior. We assess our proposed Deep SVDD models (standard Autoencoder - AE, and with Residual Connections - AEwRES), both using frozen HTS-AT embeddings. Our evaluation centers on analyzing their learned latent spaces and output anomaly scores to determine which architecture is more effective at distinguishing between normal and anomalous audio loops.

Specifically, we evaluate the models based on three criteria: (1) clear latent space separation between normal and anomalous samples, (2) informative feature representation (measured by Explained Variance Ratio), and (3) a distinct margin in the anomaly score distribution.

**Baseline Models** We benchmark against two baselines: Isolation Forest (IF) [46] and PCA-based reconstruction error. IF provides a general-purpose, non-parametric anomaly detection comparison, while PCA reconstruction error offers a simple linear alternative. This contextualizes the benefits of our deep feature learning approach for loop detection. PCA-based reconstruction error [47] evaluates whether non-linear feature learning (Deep SVDD) outperforms linear modeling. PCA is trained on normal data, and reconstruction error serves as the anomaly score, contrasting linear and non-linear approaches. All models and baselines use the same embeddings extracted by the Audio Encoder module.

**PCA Projection Visualization** We use PCA to visualize latent spaces in 2D. For Deep SVDD, PCA is applied to final latent representations; for baselines, to Audio Encoder outputs. Scatter plots show the first two components, with points color-coded as normal or anomalous based on anomaly score thresholds (95th percentile from training data). Explained Variance Ratio (EVR) is reported also.

**Latent Representation Inspection** To interpret AE and AEwRES representations, we analyze their latent dimension distributions using density histograms. These histograms show the distribution and scale of each dimension. We also generate latent activation heatmaps to visualize activation patterns and consistency across samples and latent dimensions. These visualizations offer insights into the structure and characteristics of the learned features.

---

[4] https://librosa.org/doc/latest/index.html

[5] https://wandb.ai/site/

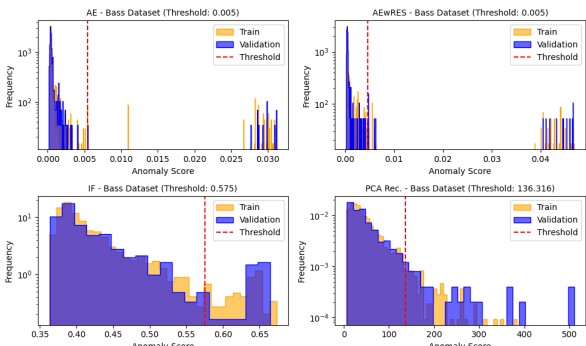

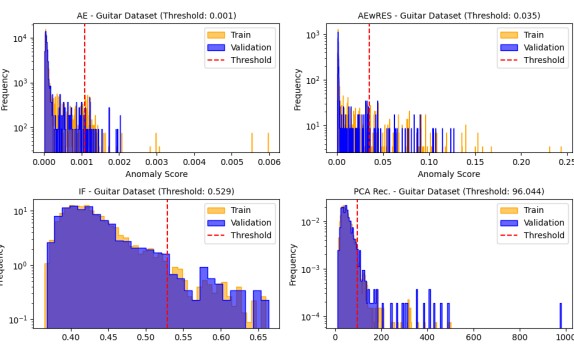

**Figure 2.** Anomaly score histograms for the bass dataset. Top row: proposed models (AE, AEwRES); bottom row: baselines (Section 5). Each plot shows score distributions for training (yellow) and validation (blue) data. Red dashed lines mark the 95th percentile anomaly threshold for training data, labeled per model: 0.005 (AE, AEwRES), 0.575 (IF), 136.316 (PCA Reconstruction).

**Figure 3.** Anomaly score histograms for Guitar dataset. Top: proposed models (AE, AEwRES); bottom: baselines (see Section 5). Each plot shows score distributions for training (yellow) and validation (blue); red dashed lines mark anomaly thresholds (95th percentile of training scores). Thresholds: AE 0.001, AEwRES 0.035, IF 0.529, PCA Reconstruction 96.044.

**Anomaly Score Distribution** We plot anomaly score distributions for all models: Deep SVDD scores use Euclidean distance from the hypersphere center; baselines use their respective metrics. Overlaid histograms (training and validation) use a log frequency scale. The 95th percentile of training scores sets the anomaly threshold, shown as a vertical line and used for color-coding and defining normal/anomalous regions.

Dimensionality reduction (PCA) and baseline modeling (IF, PCA reconstruction error) are performed using `scikit-learn`. Appendix D presents the implementation details for the PCA reconstruction error. All visualizations, including scatter plots, histograms, and heatmaps, are generated using `Matplotlib` and `Seaborn` Python libraries.

# 6 Results

We analyze Deep SVDD variants (AE, AEwRES) compared to baseline methods (IF, PCA reconstruction error) on bass and guitar datasets. Variants were selected based on preliminary experiments demonstrating the benefits of FFM and the chosen architecture. Anomalies are defined as samples with scores above the 95th percentile of the training set. Results are visualized in Figures 2, 3, 4, and Appendix F. Code and materials are available in the accompanying repository [6].

## 6.1 Performance on Bass Dataset

On the bass dataset, Deep SVDD models achieved effective separation at lower anomaly score ranges than baselines. Both AE and AEwRES produced

compact score distributions (threshold ≈ 0.005), resulting in virtually no false alarms expected at this cut-off and clear separation between normal and anomalous samples. In contrast, IF and PCA reconstruction error yielded higher and more variable scores (thresholds ≈ 0.575 and ≈ 136.3, respectively), indicating less precise normalcy definitions and potentially higher false positive rates. AEwRES yielded tightly clustered latent representations, with sharp KDE peaks (density ≈ 120), clear PCA separation, and high PC1 variance (89.6%), indicating better capture of normal bass patterns. AE showed broader distributions (peak density ≈ 60), lower variance (74.7%), and less distinct clustering, suggesting a less distinctive latent space than AEwRES. Box plots (Figure 4) confirm that AEwRES and AE assign lower scores to normal data than baselines, with AEwRES showing the most compact distribution and strongest outlier separation.

## 6.2 Performance on Guitar Dataset

On the guitar dataset, the AE model produced tightly clustered scores (below 0.001, threshold ≈ 0.001), while AEwRES better captured the data's diversity, yielding a broader score range (0–0.2, threshold ≈ 0.035). AEwRES's latent space exhibited a distinct structure: KDE plots revealed multiple peaks, which can be interpreted as diverse playing styles or techniques captured by the model. For instance, these peaks might correspond to different musical motifs, such as variations in strumming patterns or shifts between finger-style riffs and chord-based progressions. In essence, AEwRES is sensitive to subtle nuances in the performance, adding depth to its anomaly detection capabilities. PCA projections isolated normal data from anomalies. AEwRES explained 93.2% of variance in the first two princi-

---

[6] https://github.com/dadmaan/music-anomalizer.git

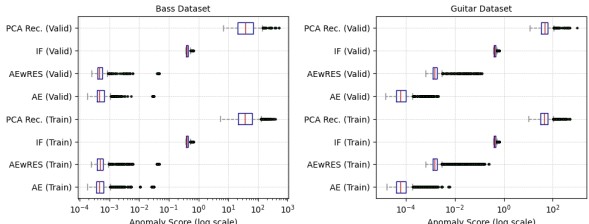

**Figure 4.** Box plots of anomaly score distributions for AE, AEwRES, and baselines (Section 5) on Bass (left) and Guitar (right) datasets. Eight vertical box plots per panel show training and validation data for each model. Scores are shown on a logarithmic scale, summarizing medians, quartiles, and outliers.

pal components (PC1=70.1%, PC2=23.1%), versus 77.7% for AE, indicating richer latent representations. Additionally, AEwRES heatmaps visualized diverse, structured normal patterns, whereas AE's space was over-compressed, with sharply peaked KDEs (peak ≈ 160) and less separation in PCA, suggesting that AE may miss subtle anomalies. Both Deep SVDD models outperformed the baselines, which struggled with the diversity of the guitar data and exhibited overlap between normal and anomalous representations (Appendix F). AEwRES, in particular, maintained improved separation, with structured latent space and effective thresholding, demonstrating superior suitability for the guitar dataset.

## 6.3 Compartive Performance Summary

Choose AEwRES when data variability is high. Deep SVDD models (AE, AEwRES) outperform IF and PCA baselines across datasets, yielding lower anomaly scores for normal data and clearer separation at detection thresholds (Figure 4). While AE achieves high compactness on simpler data (bass), it tends to over-compress on diverse datasets (guitar). AEwRES enhances feature separability and generalization, effectively modeling complex patterns while maintaining separation, particularly on the guitar dataset (Appendix F). Thus, AEwRES is preferable for loop detection in varied musical contexts due to its better separation and ability to capture pattern diversity.

## 7 Discussion

We evaluated automated loop detection using Deep SVDD with our AEwRES encoder (integrating FFM and HTS-AT), benchmarking against IF and PCA-based reconstruction error (Section 5). This comparison assessed the benefits of deep feature learning versus simpler methods. To clearly demonstrate

our contribution, we emphasized not just raw accuracy but also evaluated how well these models captured meaningful musical structures, which are vital in understanding and reinforcing temporal and hierarchical patterns in music. This highlights the novelty of our approach, as capturing these nuances contributes to more effective loop detection. To isolate the impact of residual connections, we also compared AEwRES with a standard Autoencoder (AE) within the Deep SVDD framework. On both bass and guitar datasets, AEwRES outperformed baselines (IF, PCA reconstruction error) and the AE variant, demonstrating effective feature learning for loop detection. Integrating FFM with HTS-AT overcomes fixed-length input limitations (Section 2), with FFM effectively aggregating temporal information for HTS-AT to capture local and global dependencies (Appendix E).

Performance varied between bass and guitar datasets, indicating adaptability to different music collections. In the realm of signal processing, a structured representation often provides a clearer pathway for feature identification compared to merely compact representations. On bass, both AE and AEwRES learned compact representations, but AEwRES produced a more structured latent space. This structural advantage became more apparent on the guitar dataset, where AE tended to over-compress, a condition where essential details might be lost in favor of reducing dimensionality, resulting in lower explained variance (77.7%) compared to AEwRES (93.2%). In essence, while compactness seeks to reduce redundancy, it can inadvertently discard valuable contextual information. Indeed, AEwRES adapted well to the guitar dataset by modeling its variability with a structured latent space, albeit a broader latent representation, as evidenced by PCA plots and explained variance metrics. This enabled Deep SVDD to define normality more effectively, suggesting that such architectures, incorporating elements such as residual connections, are well-suited for complex and diverse musical audio.

Contextualizing these results, our application of Deep SVDD directly to audio features encoded by HTS-AT supports our hypothesis that such models can form the basis for loop detection and analysis, particularly when designed to capture nuanced temporal and hierarchical information. Qualitatively, samples flagged as anomalous often contained transient irregularities (e.g., fret noise, clicks) or harmonic structures that deviated significantly from the standard progressions in the training set. The effectiveness of the learned representations aligns with findings that emphasize the practicality of pretrained foundation models for audio tasks [14, 19]. The AEwRES architecture's performance echoes the benefits seen from residual connections [29] in other domains.

## 7.1 Possible Applications

The proposed method enables automated quality control and creative discovery in music production. For quality control, it flags loops with artifacts (e.g., clicks, phase issues, noise), reducing manual auditing. Its unsupervised design removes the need for labeled anomalies, with performance depending on training data representativeness. Creatively, anomaly detection surfaces unusual loops for experimental sound design, aligning with serendipitous retrieval [48, 49]. Producers can use high anomaly scores to identify samples with atypical rhythms or textures, aiding exploration.

Beyond these applications, the approach also holds potential for integration with Digital Audio Workstations (DAWs), where real-time loop anomaly detection could alert artists to problematic loops (e.g., timing errors, unwanted noise) before production. A more speculative application involves style transfer and genre adaptation. By training the model on a music collection of a specific genre (e.g., jazz guitar), users could detect deviations that signal potential for cross-genre influence. Moreover, the hybrid approach—utilizing deep audio representations with one-class learning—eliminates the need for large, labeled datasets, thereby supporting individuals with limited resources. Ultimately, this functions as an assistant that flags deviations for review, leaving the final subjective judgment of whether an artifact is a 'flaw' or a 'creative feature' to the human producer.

## 8 Limitations

Several limitations should be noted. Our evaluation uses unsupervised anomaly detection without ground-truth labels, so anomalies are defined heuristically via a 95th percentile threshold, which may not be optimal or consistent across data. To address this issue, introducing a small, curated set of proxy-labeled or synthetic anomalies could help in calibrating the 95th percentile threshold, making our claims more verifiable. This validation set could provide an evidence-based metric, allowing us to report its precision-recall scores, thus transforming the current heuristic approach into a more robust evaluation strategy. The interpretation of anomalies is also subjective and context-dependent. Our experiments focus only on bass and guitar loops from a specific dataset, limiting generalizability to other instruments, genres, or audio types. The dataset's diversity may not reflect real-world audio encountered by users. Methodologically, Deep SVDD assumes normal data can be enclosed within a single hypersphere in latent space, which may not capture complex, multi-modal distributions. Using a deep encoder mitigates this, but the HTS-AT architecture increases computational demands. Dataset biases, such as prevalent playing styles or recording qualities, may also affect learned representations and detection performance.

## 9 Future Work

Building upon these findings and limitations, future research could proceed in several directions. Thematic areas such as generalizability, efficiency, and usability can serve as anchors for these directions. To enhance generalizability, it is essential to expand the assessment to include diverse instruments, genres, and audio formats. This will help evaluate the applicability of the AEwRES-Deep SVDD method across various contexts. Incorporating partially annotated datasets or semi-supervised techniques can enable more rigorous quantitative evaluation and threshold refinement, thereby supporting a more robust generalization. In terms of efficiency, fine-tuning the HTS-AT model for loop detection or leveraging knowledge distillation could lead to the development of lighter, real-time-capable models. For usability, improvements in workflow integration, such as combining loop detection with tools like transcription and source separation, could refine audio analysis by honing in on anomalous segments. Additionally, enhancing the user interface by introducing elements for sensitivity adjustment and facilitating practitioner feedback will boost practical utility.

## 10 Conclusion

In conclusion, this work demonstrated that Deep SVDD, coupled with an audio encoder like HTS-AT with FFM, offers a viable approach for loop detection. The AEwRES variant, in particular, showed promise due to its ability to learn discriminative latent representations that accommodate the diversity inherent in complex musical data, compared to the selected baselines (IF and PCA reconstruction error) and the AE variant. While limitations exist regarding the evaluation methodology and dataset scope, the results indicate the potential of this approach to alleviate challenges in automated audio analysis. By providing a means to automatically identify deviations from normative patterns directly within the audio domain and handling variable-length inputs effectively, this research lays the groundwork for future investigations into more nuanced, interpretable, and widely applicable music analysis systems, as elaborated in Section 7.1.

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

# A  System Architecture

Figure A.1 depicts the proposed loop detection model, which couples an Audio Encoder for hierarchical feature extraction with a Deep SVDD module for anomaly detection. The Audio Encoder employs duration-aware dual pathways with Mel-FilterBank processing, Conv2D layers, a feature-fusion mechanism, and a pre-trained HTS-AT backbone. The Deep SVDD comprises five encoder layers that project features into a latent space, mapping normal samples within a hypersphere and abnormal samples outside. The system supports end-to-end, unsupervised training. It produces distance-based anomaly scores from the learned representation. The Audio Encoder illustration is inspired by Wu et al. [19].

**Table B.1.** Statistics per instrument of the curated dataset

| | Instrument | Bass | Guitar |
|---|---|---|---|
| | Count | 1816 | 4294 |
| | Durations (hrs) | 2.337 | 5.924 |
| BPM | mean | 111.779 | 108.785 |
| | std | 18.854 | 18.000 |
| | min | 64.000 | 73.000 |
| | 25% | 95.000 | 95.000 |
| | 50% | 114.000 | 110.000 |
| | 75% | 123.000 | 120.000 |
| | max | 170.000 | 170.000 |
| Duration (secs) | mean | 4.633 | 4.967 |
| | std | 2.436 | 1.971 |
| | min | 0.546 | 0.417 |
| | 25% | 3.692 | 4.000 |
| | 50% | 4.364 | 4.571 |
| | 75% | 5.333 | 5.647 |
| | max | 28.346 | 22.700 |

# B  Dataset Statistics

Table B.1 summarizes a curated set of 1,816 bass and 4,294 guitar samples totaling 2.337 h and 5.924 h, respectively. Tempi cover moderate ranges (bass: 64–170 BPM, mean 111.779; guitar: 73–170 BPM, mean 108.785), with interquartile spans of 95–123 BPM (bass) and 95–120 BPM (guitar). Clips are short (mean durations 4.633 seconds for bass and 4.967 seconds for guitar). As illustrated in Fig. B.1, samples are distributed across musical genres and keys.

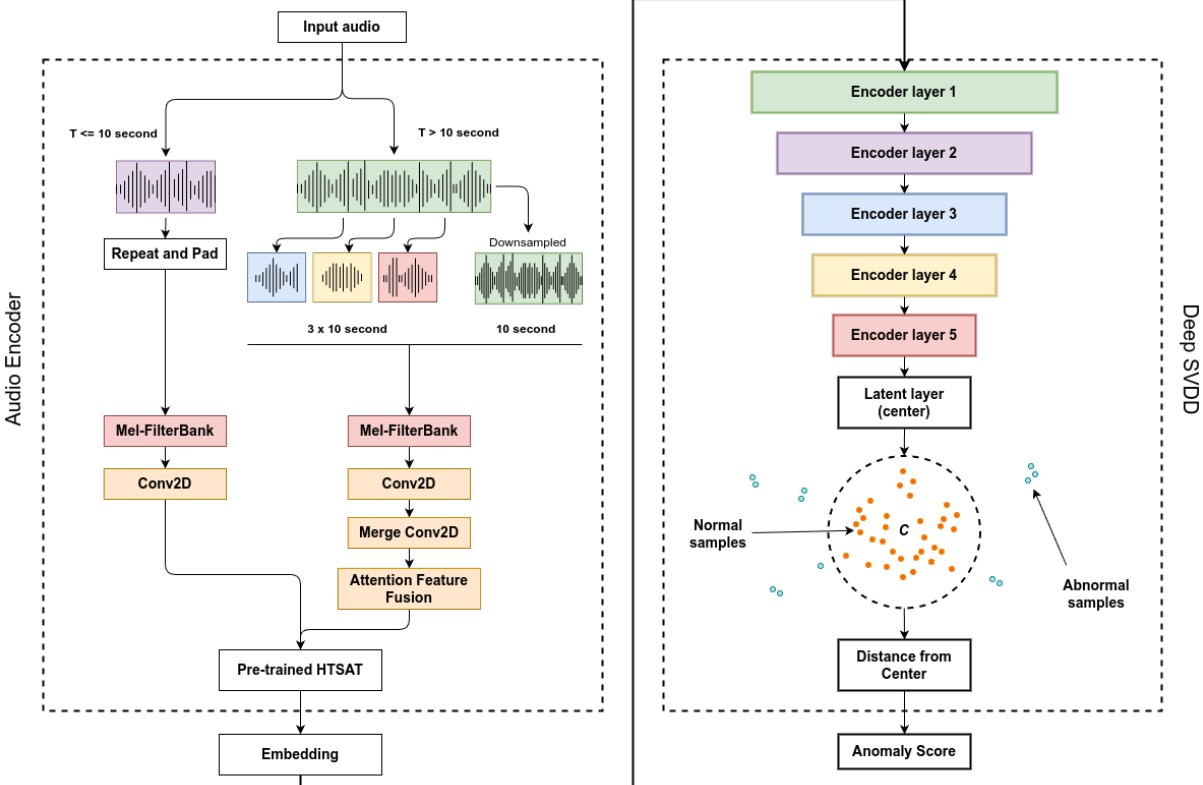

**Figure A.1.** Architectural overview of the proposed loop detection model. The model comprises two main components: (left) an Audio Encoder module for hierarchical feature extraction from audio inputs and (right) a Deep SVDD module for anomaly detection. The Audio Encoder processes input audio through dual pathways based on duration, utilizing Mel-FilterBank processing, Conv2D layers, the feature fusion mechanism, and a pre-trained HTS-AT for feature extraction. The Deep SVDD module consists of five encoder layers that transform features into a latent space, where normal samples (orange dots) are mapped within a hypersphere and abnormal samples (blue dots) are mapped outside. The architecture enables end-to-end training for unsupervised loop detection and anomaly identification through distance-based scoring from the learned latent representation. The illustration of the Audio Encoder is adapted from Wu et al. [19].

## C   AutoEncoder Architecture

Figure C.1 presents the AEwRES autoencoder architecture inspired by U-Net, comprising a five-layer encoder and five-layer decoder operating on 1024-dimensional inputs. The encoder progressively reduces dimensionality, which is then reconstructed by the decoder. Residual connections (dotted) add the output of encoder layer i to the input of decoder layer 5-i; the baseline AE shares the same architecture but omits these residual links.

## D   PCA Reconstruction Error Algorithm

Principal Component Analysis (PCA) based reconstruction error is a widely used technique for unsupervised anomaly detection [51, 52]. It operates under the assumption that the majority of the training data represents normal behavior, and that this normal data lies predominantly within a lower-dimensional subspace of the original feature space. Anomalies, conversely, are expected to deviate from this normal subspace. PCA is employed to identify this principal subspace from the training data. The anomaly score for any given data point is then calculated as the error incurred when attempting to reconstruct the point after projecting it onto this learned normal subspace. Points that deviate substantially from the normal patterns captured by the principal components will exhibit a high reconstruction error.

The complete implementation procedure is formally described in Algorithm D.1.

**Algorithm D.1** PCA Reconstruction Error for Anomaly Detection

---

**Require:** Training data embeddings $\mathbf{X}_{\text{train}} \in \mathbb{R}^{n_{\text{train}} \times d}$
**Require:** Evaluation data embeddings $\mathbf{X}_{\text{eval}} \in \mathbb{R}^{n_{\text{eval}} \times d}$
**Require:** Number of components $n_{\text{components}}$ (integer, float, or None)
**Require:** Variance threshold $\theta_{\text{var}}$ (default 0.95)
**Require:** `standardize` (boolean)
1: Let $\mathbf{X}'_{\text{train}} = \mathbf{X}_{\text{train}}$ and $\mathbf{X}'_{\text{eval}} = \mathbf{X}_{\text{eval}}$
2: **if** standardize is True **then**
3:      Compute mean $\boldsymbol{\mu}$ and standard deviation $\boldsymbol{\sigma}$ from $\mathbf{X}_{\text{train}}$
4:      Standardize: $\mathbf{X}'_{\text{train}} \leftarrow (\mathbf{X}_{\text{train}} - \boldsymbol{\mu})/\boldsymbol{\sigma}$
5:      Standardize: $\mathbf{X}'_{\text{eval}} \leftarrow (\mathbf{X}_{\text{eval}} - \boldsymbol{\mu})/\boldsymbol{\sigma}$
6: **end if**
7: **if** $n_{\text{components}}$ is None **then**
8:      Fit PCA on $\mathbf{X}'_{\text{train}}$ (full rank)
9:      Compute cumulative explained variance ratios
10:      $k \leftarrow \min\{j : \sum_{i=1}^{j} \text{variance}_i \geq \theta_{\text{var}}\}$
11:      $n_{\text{selected}} \leftarrow k$
12: **else if** $n_{\text{components}}$ is float and $0 < n_{\text{components}} < 1$ **then**
13:      $n_{\text{selected}} \leftarrow n_{\text{components}}$             ▷ as explained variance
14: **else**
15:      $n_{\text{selected}} \leftarrow n_{\text{components}}$             ▷ as integer
16: **end if**
17: Fit PCA model on $\mathbf{X}'_{\text{train}}$ with $n_{\text{selected}}$ components
18: Initialize empty lists $\mathbf{e}_{\text{train}}$ and $\mathbf{e}_{\text{eval}}$
19: **for** each $\mathbf{x}$ in $\mathbf{X}'_{\text{train}}$ **do**             ▷ Calculate training errors
20:      Project: $\mathbf{z} \leftarrow \text{PCA\_TRANSFORM}(\mathbf{x})$
21:      Reconstruct: $\hat{\mathbf{x}} \leftarrow \text{PCA\_INVERSE\_TRANSFORM}(\mathbf{z})$
22:      Compute error: $e \leftarrow \sum_{i=1}^{d}(x_i - \hat{x}_i)^2$
23:      Append $e$ to $\mathbf{e}_{\text{train}}$
24: **end for**
25: **for** each $\mathbf{x}$ in $\mathbf{X}'_{\text{eval}}$ **do**            ▷ Calculate evaluation errors
26:      Project: $\mathbf{z} \leftarrow \text{PCA\_TRANSFORM}(\mathbf{x})$
27:      Reconstruct: $\hat{\mathbf{x}} \leftarrow \text{PCA\_INVERSE\_TRANSFORM}(\mathbf{z})$
28:      Compute error: $e \leftarrow \sum_{i=1}^{d}(x_i - \hat{x}_i)^2$
29:      Append $e$ to $\mathbf{e}_{\text{eval}}$
30: **end for**
31: **return** array of training errors $\mathbf{e}_{\text{train}}$, array of evaluation errors $\mathbf{e}_{\text{eval}}$

---

# E   Preliminary Experiments

This appendix details preliminary experiments conducted to validate key design choices and select promising model architectures for the main evaluation presented in Section 6. To facilitate rapid iteration and efficient hyperparameter exploration, these initial tests were performed on a smaller, representative subset of the bass dataset (described in Section 4.1), comprising 392 bass loops from the MusicRadar catalog [7]. The primary objectives were to:

1. Evaluate the effectiveness of the FFM described in our methodology (Section 3).

2. Compare different network architectures to se-

lect the most promising candidates for the subsequent evaluation in Section 6.

First, we assessed the impact of incorporating the FFM by comparing model variants ($AE$ and $AEwRES$) trained with and without it. The results demonstrated the benefits of FFM. Models utilizing FFM converged faster and achieved improved representational quality. Specifically, FFM facilitated the models to capture underlying patterns, as evidenced by latent space distributions (KDE plots) and tighter clustering of normal samples in PCA projections (Figure E.2) and clearer separation in anomaly score distributions (Figure E.1) compared to models without it. Consequently, FFM was adopted for subsequent architecture comparisons and main experiments (Section 6).

Following FFM validation, hyperparameter tuning was conducted for $AE$ and $AEwRES$ architectures

---

[7] https://www.musicradar.com/news/tech/sampleradar-392-free-bass-guitar-samples-537264

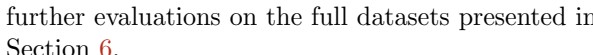

further evaluations on the full datasets presented in Section 6.

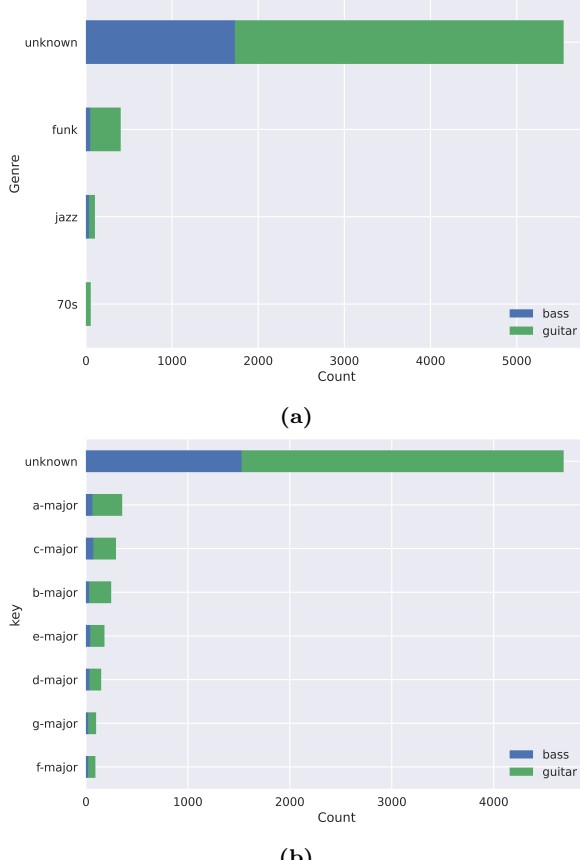

**(a)**

**(b)**

**Figure B.1.** Distribution of curated guitar and bass samples across (a) musical genres and (b) musical keys. The height of the bars indicates the count of samples for each category, with colors differentiating between bass (blue) and guitar (green). The prevalence of the 'unknown' category highlights common challenges with sample library metadata.

to optimize performance. The tuned $AE$ exhibited a negatively skewed latent distribution ranging from -1.0 to 0.25, while $AEwRES$ showed a more balanced, symmetrical distribution centered around zero (-0.4 to 0.4). Feature representation heatmaps revealed higher contrast in $AEwRES$, suggesting more distinct feature capture. Anomaly detection performance indicated successful separation of normal samples and potential anomalies in both architectures; however, $AEwRES$ achieved a lower anomaly threshold (q=0.95 $\approx$ 0.009) compared to $AE$ ($\approx$ 0.016), potentially indicating enhanced precision. During tuning, $AEwRES$ generally demonstrated faster convergence and more stable training dynamics.

In summary, the optimized $AEwRES$ configuration showed advantages on this subset in terms of representation balance, feature distinctiveness, potential precision, and training efficiency. Nonetheless, given that $AE$ also performed competently after tuning and represents a different architectural approach, we selected both configurations for the

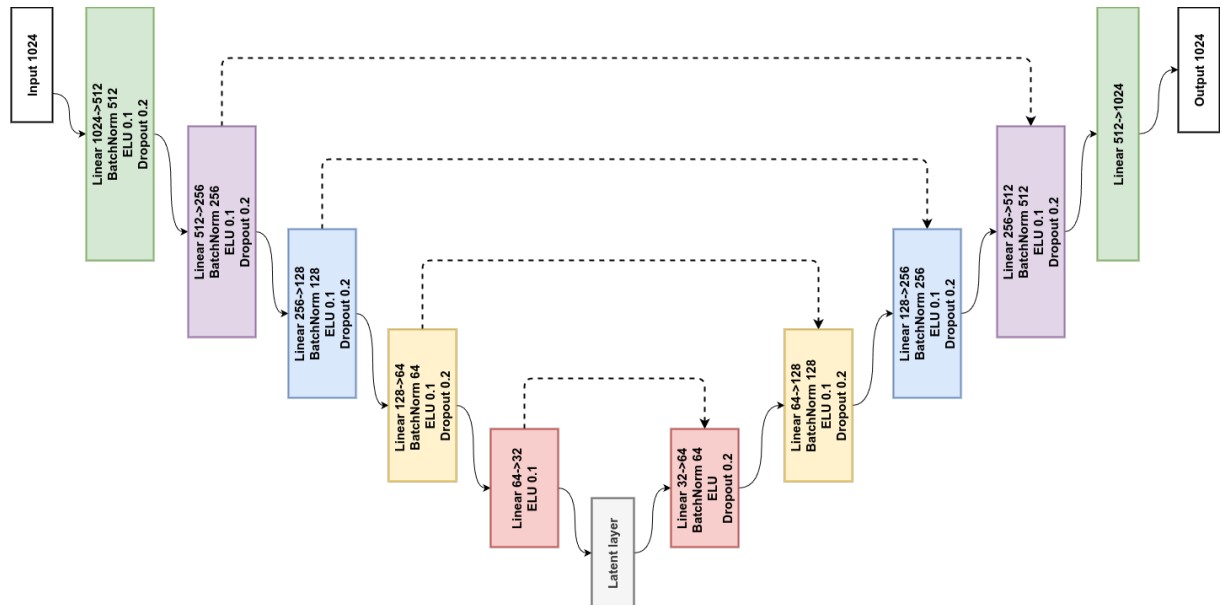

**Figure C.1.** The architecture of the autoencoder (*AE*) network with residual connections (dotted arrows) (*AEwRES*). The model follows a symmetric design (inspired by U-net architecture [50]) with an encoder (left) and decoder (right). The input dimension of 1024 is progressively reduced through five encoder layers and then reconstructed through five decoder layers. Each layer comprises a linear transformation followed by BatchNorm, ELU activation (*alpha*=0.1), and Dropout (rate=0.2). Residual connections add the output of encoder layer $i$ to the input of decoder layer $5 - i$. Note that the base *AE* architecture is identical to that of *AEwRES*, but without any residual connections.

# F Main Experiment Supplementary Materials

This appendix provides additional visualizations for the results presented in Section 6. Figures F.1 and F.2 presented in this appendix illustrate the performance of the models on the bass and guitar datasets, respectively. They provide a more detailed analysis of the results, discussed in Section 6. The figures show the density distributions of latent values across multiple dimensions, PCA scatter plots, and heatmaps of latent space representations, for training and vlidation phases. These visualizations are useful for understanding the behavior of the models and how they capture the underlying patterns in the data.

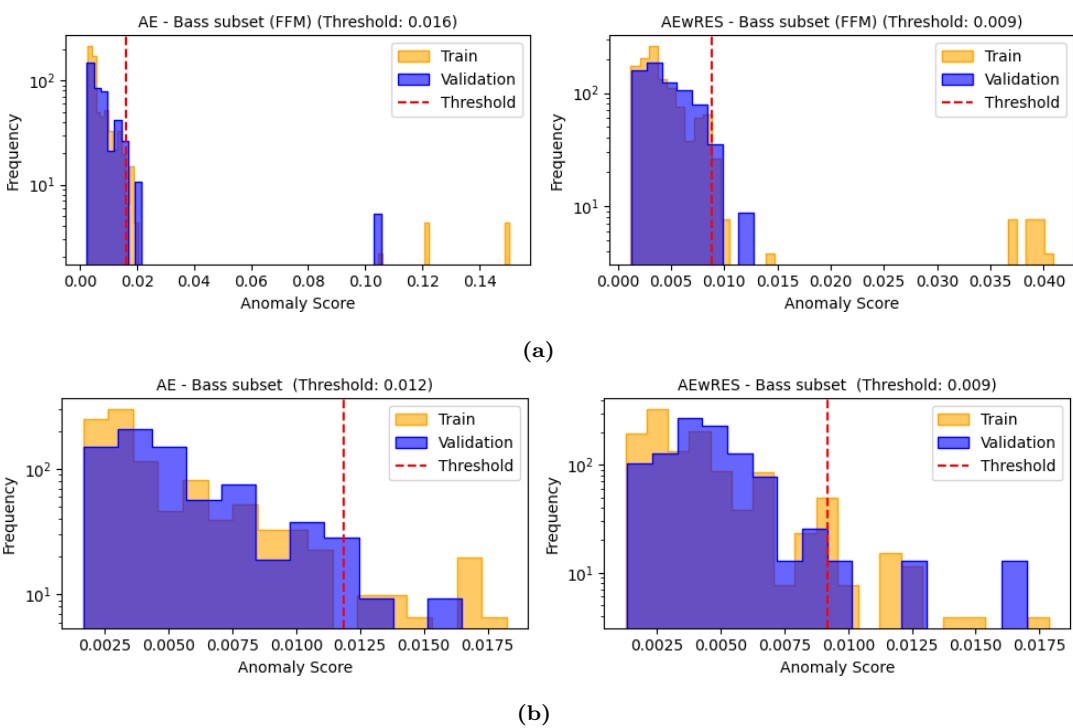

**Figure E.1.** The figure displays anomaly score distribution histograms for *AE* and *AEwRES* applied to the subset of bass dataset (392 samples). The top row (a) shows the scores obtained with FFM and the bottom row (b) displays the scores obtained witout FFM. Each plot depicts the frequency distribution of anomaly scores for both training data (yellow) and validation data (blue), with red dashed lines indicating the anomaly threshold set at the 95th percentile of the training scores.

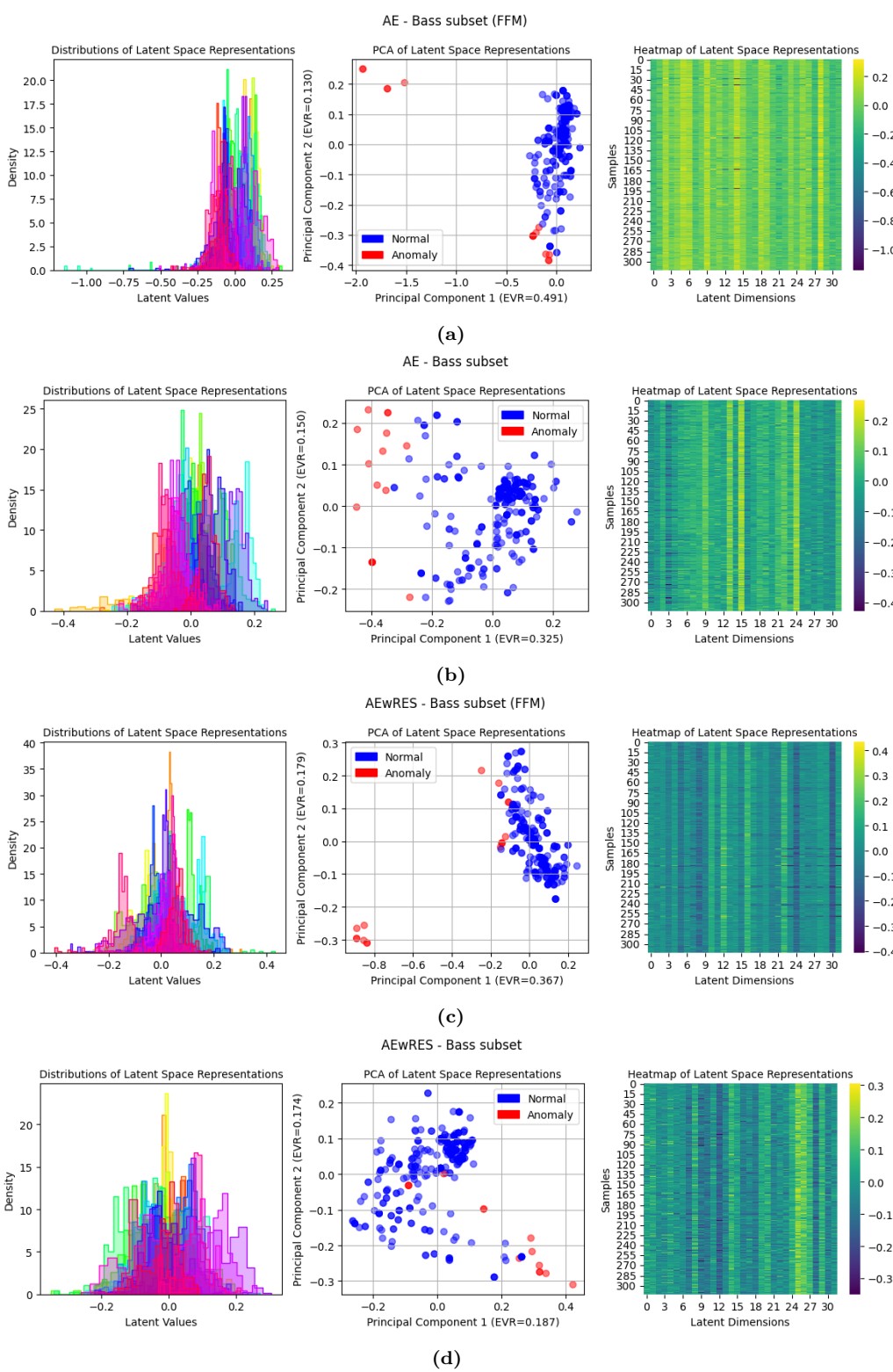

**Figure E.2.** The figure presents three visualizations of latent space representations from *AE* and *AEwRES* models applied to the subset of bass dataset (392 samples). The first and third rows (a, c) show the results obtained with FFM and the second and fourth rows (b, d) display the results witout FFM. For each row, the left panel displays density distributions of latent values across multiple dimensions. The center panel shows a PCA scatter plot projecting the latent space onto two principal components, with blue points representing normal samples and red points indicating anomalies. The right panel features a heatmap of latent space representations across 32 dimensions (x-axis) for $N$ samples (y-axis), with color intensity reflecting latent values.

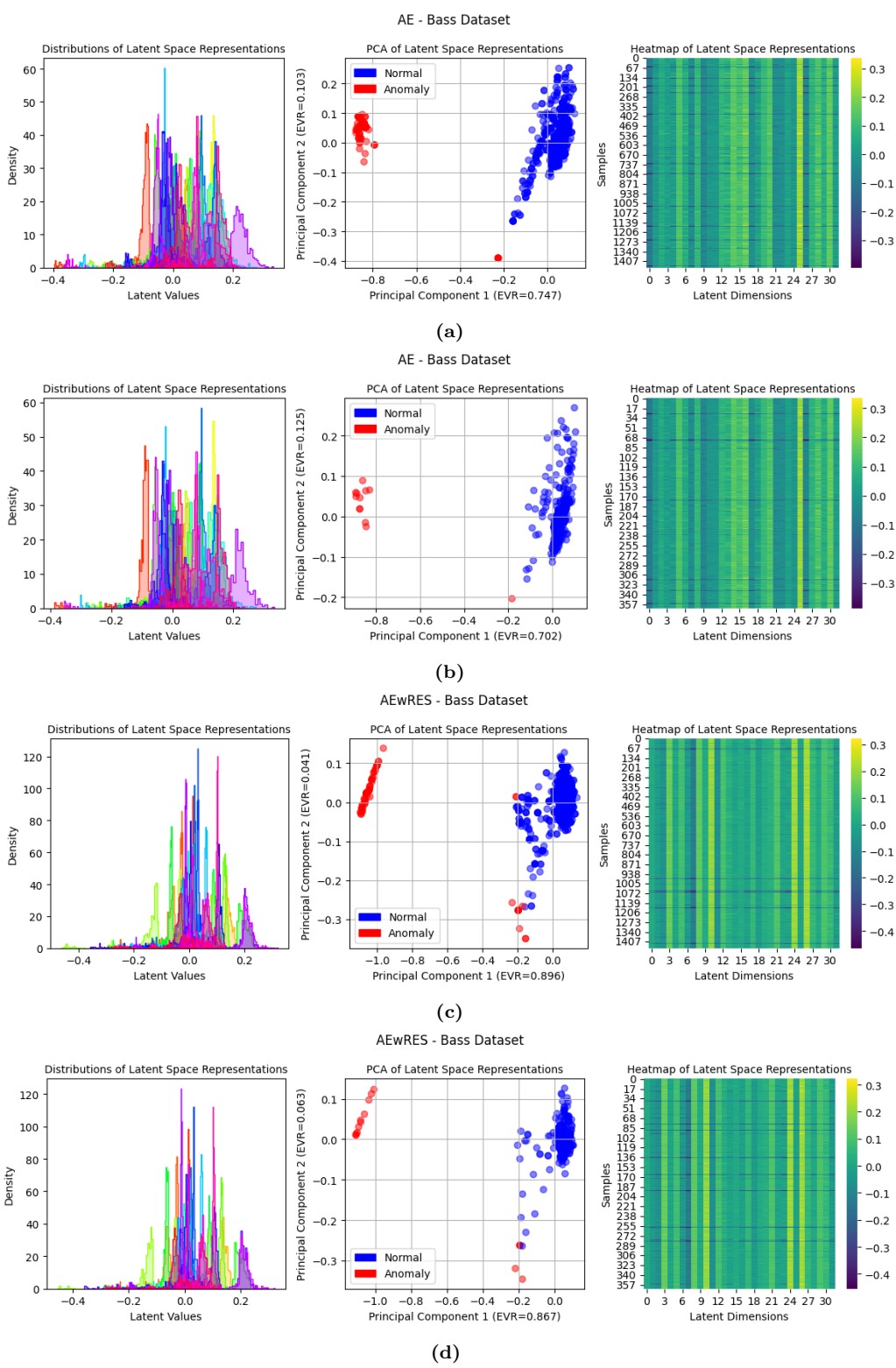

**Figure F.1.** Comparative analysis of (a,b) *AE* and (c,d) *AEwRES* models using bass dataset, as described in Section 4.1. The first two rows (a,b) show the results obtained by *AE* model during (a) training and (b) validation phases. The second two rows (c,d) displays the results obtained by *AEwRES* model during (c) training and (d) validation phases. For each row, the left panel displays density distributions of latent values across multiple dimensions. The center panel shows a PCA scatter plot projecting the latent space onto two principal components, with blue points representing normal samples and red points indicating anomalies. The right panel features a heatmap of latent space representations across 32 dimensions (x-axis) for $N$ samples (y-axis), with color intensity reflecting latent values.

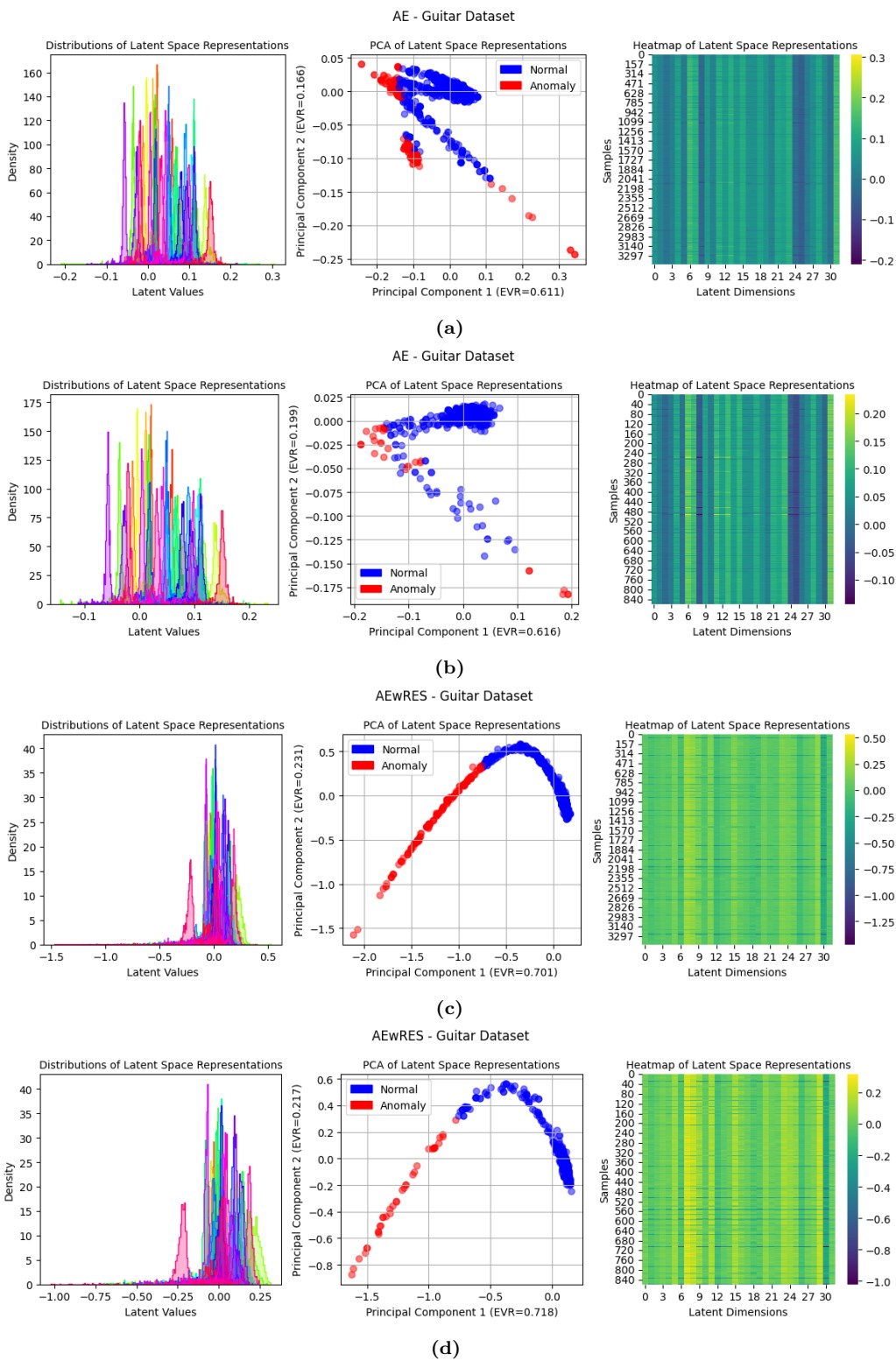

**Figure F.2.** Comparative analysis of (a,b) *AE* and (c,d) *AEwRES* models using guitar dataset, as described in Section 4.1. The first two rows (a,b) show the results obtained by *AE* model during (a) training and (b) validatio phases. The second two rows (c,d) displays the results obtained by *AEwRES* model during (c) training and (d) validatio phases. For each row, the left panel displays density distributions of latent values across multiple dimensions. The center panel shows a PCA scatter plot projecting the latent space onto two principal components, with blue points representing normal samples and red points indicating anomalies. The right panel features a heatmap of latent space representations across 32 dimensions (x-axis) for $N$ samples (y-axis), with color intensity reflecting latent values.

