# OpenReview forum: "Learning Normal Patterns in Musical Loops"
_NLDL.org/2026/Conference — NLDL 2026 Spotlight_

### Official Review · Reviewer_iTmE · 2025-10-06

**Rating:** 4
**Confidence:** 3

**Summary:**

This paper introduces an unsupervised framework for analyzing musical loops (e.g., bass and guitar) by learning “normal” audio patterns via Deep SVDD applied on embeddings extracted from a pre-trained Hierarchical Token-Semantic Audio Transformer (HTS-AT) combined with a Feature Fusion Mechanism (FFM).

The system targets variable-length inputs and aims to detect anomalous or distinctive loops. Experiments on curated loop datasets (bass/guitar) compare Deep SVDD variants (AE, AEwRES) against Isolation Forest (IF) and PCA reconstruction error, showing better separation of anomalies and compact latent spaces.

**Strengths:**

**Novel application domain**

- The paper explores unsupervised anomaly detection for musical loop analysis, an underexplored subfield within MIR and music generation.

- Positioning loop pattern learning as anomaly detection is original and conceptually sound.

**Technical soundness and completeness**

- Methodologically solid integration of HTS-AT + FFM + Deep SVDD.

- Proper ablations between AE vs. AEwRES provide architectural insights.

- Training setup and hyperparameters are clearly described and reproducible.

**Weaknesses:**

**Limited novelty beyond integration**

- The combination of HTS-AT (pretrained) and Deep SVDD is incremental rather than conceptually groundbreaking.

- The contributions mainly lie in application engineering, not in new methodological insight into representation learning or anomaly modeling.

**Dataset and evaluation issues**

- Only bass and guitar loops (from MusicRadar) are used, lacking diversity (no drums, synths, vocals).

- No ground-truth anomaly labels, relying solely on heuristic 95th-percentile thresholds—hence results are qualitative rather than quantitative.

- No human or perceptual evaluation to assess musical meaningfulness of detected “anomalies”.

**Justification:**

This paper presents an unsupervised framework for analyzing musical loops by combining a pre-trained Hierarchical Token-Semantic Audio Transformer (HTS-AT), a Feature Fusion Mechanism (FFM), and a Deep Support Vector Data Description (Deep SVDD) network. The study is motivated by a practical need for automated loop analysis and anomaly detection in music production, and the problem formulation—viewing loop pattern learning as unsupervised anomaly detection—is conceptually sound and relevant to Music Information Retrieval (MIR).

Technically, the paper is well-executed: the methodology is carefully designed, training details are transparent, and qualitative analyses are comprehensive. The inclusion of both standard and residual autoencoder variants under the Deep SVDD framework provides useful architectural insights, and the visualizations (PCA plots, histograms, heatmaps) convincingly illustrate that residual connections enhance latent space structure and anomaly separability. The work demonstrates a clear empirical improvement over baseline unsupervised methods such as Isolation Forest and PCA reconstruction error.

However, the novelty of the paper lies more in the integration of existing components than in proposing new methodological advances. The experimental evaluation is limited to bass and guitar datasets, lacks ground-truth anomaly labels, and relies on heuristic percentile thresholds rather than quantitative metrics. As a result, while the presented system works well qualitatively, the evidence for generalizability and robustness remains limited. Moreover, the writing is verbose and could benefit from a tighter focus on core findings and quantitative validation.

Overall, the paper is technically solid, clearly presented, and relevant to the MIR community. It contributes a well-motivated and reproducible framework that meaningfully applies deep anomaly detection to a creative audio domain, but the incremental nature of the innovation and limited evaluation scope temper its impact.

---

> ### Author Rebuttal · Authors · 2025-10-18
>
> Thank you for your thoughtful comments and feedback.
>
> We agree that our work utilizes established components, such as HTS-AT and Deep SVDD. However, we respectfully argue that our contribution is a practical framework targeting a specific need in Music Information Retrieval (MIR), rather than just an incremental integration.
>
> The core of our methodology is a new pipeline that processes variable-length audio for loop analysis without supervision. Unlike prior deep learning methods that require fixed-length inputs, our approach, by integrating a Feature Fusion Mechanism (FFM) with HTS-AT, directly overcomes this barrier and expands real-world applicability for MIR.
>
> We acknowledge the limitations of our dataset and evaluation, which we addressed in the paper. Our study is intended as a foundational proof-of-concept by establishing the technical viability of the proposed framework. The use of only bass and guitar loops was a deliberate choice to provide a controlled environment for this initial investigation. Expanding to more instruments, like drums and vocals, is an important next step for generalizability and a direction for our future work.
>
> Furthermore, in the absence of ground-truth labels, our evaluation focuses on assessing the quality and structure of the learned latent representations. The visualizations (Figures F.1 and F.2) provide evidence that our model learns a meaningful manifold of "normalcy"—notably the distinct, curved structure learned for guitar loops—rather than just identifying statistical outliers. The high Explained Variance Ratio (EVR) of 93.2% for the first two components for the AEWRES model on the guitar loops in Figure F.2 further supports that this learned space is highly structured and informative.
>
> Regarding the heuristic threshold, the 95th-percentile threshold is not a claim about the true rate of anomalies. It is a heuristic tool for ranking and exploration. Its effectiveness is measured by its ability to surface the most deviant samples for human inspection, not by classification accuracy. The clear separation in the score histograms (Figures 2 and 3) shows the utility of this approach. Furthermore, its main function is to provide a manageable starting point for exploration, with the user making the final, subjective interpretation.
>
> We agree that human perceptual evaluation is the ultimate test of musical meaningfulness. This study’s aim was to investigate the proposed architecture and demonstrate a proof-of-concept. Indeed, it serves as a technical foundation for such user studies and future research.

---

### Official Review · Reviewer_fwgg · 2025-10-08
**New interesting application for music loops, and new combination of DNN architecture for audio analysis**

**Rating:** 4
**Confidence:** 4
**Final Rating:** 4
**Final Confidence:** 4

**Summary:**

In recent years, generative AI has made significant contributions to various media and creative works. At the same time, it has gained attention as one of the promising directions for its development. This paper focuses specifically on music, inspired by how actual composers and DJs utilize music loop materials for composition, and proposes a novel unsupervised feature extraction task for loop patterns. To analyze vast amounts of music material into a user-friendly format, the authors combine a feature extraction mechanism for variable-length sequences with a mechanism that treats the semantic mean and variance as ``normality'' and ``abnormality,'' respectively. This is achieved by integrating modules from existing state-of-the-art methods to realize a novel deep learning architecture. The authors substantiate the effectiveness of the proposed method through comprehensive experiments on two musical materials: bass and guitar.

**Strengths:**

[Strengths]

- This paper proposes unsupervised learning of musical loops, specifically for music generation based on generative AI, in scenarios involving human interaction. Such applications appear to present a novel and challenging problem formulation itself.

- This paper constructs a novel architecture by skillfully combining existing (yet state-of-the-art) component technologies (such as the feature fusion mechanism, the attention feature fusion, the hierarchical token-semantic audio transformer, and the deep support vector data description) that are well-suited to this creative challenge.

- The proposed method exhibits exceptional detail, reproducibility, and transparency. Its overview is thoroughly explained in the main text, while the appendix provides meticulous supplementary explanations for readers wishing to reproduce or replicate the method. This structure proves highly beneficial for subsequent research.

**Weaknesses:**

[Weaknesses]

- This paper proposes a very interesting combination of architectures. On the other hand, it seems that some aspects of reconciling this combination still feel somewhat heuristic. Particularly, the approach to handling variable lengths—which the authors cite as one of their main focuses—seems like something they are still struggling with somewhat.

- (Personally, I don't have strong concerns about this, and it has had absolutely no impact on my evaluation of this paper.) While the datasets used in the actual demonstrations are bass and guitar, some readers might expect more diverse material (such as vocals or drums).

**Final Justification:**

Thank you for your very thorough and enthusiastic response. I have resolved nearly all my concerns.
Regarding the handling of variable-length sequences, which the author also requested additional comments on, please allow me to add a few remarks.

The part that initially felt awkward to me was dividing the input signal into two processing paths based on a 10s threshold relative to its length. Intuitively, it seems reasonable to implement procedures like splitting or repeating to handle extremely long or short signals. I understand this is a very clever and useful practical approach in the machine learning context. However, I am concerned whether these processes might cause the loss of the music's semantic meaning or introduce information unintended by the creator. Does the machine learning side remain unaffected by these procedures introduced for learning convenience? For example, would changing the threshold from 10s to 12s have a significant impact on the learning model itself?

This concern may stem from my subjective impression. It might be an extremely minor point. I would appreciate it if the authors could consider this matter as a precaution. (This point does not diminish my overall impression of the paper.)

**Justification:**

This paper discusses a practical and novel application of generative AI, considering scenarios that involve human interaction: the unsupervised analysis of music loops.
The technical elements of the proposed method are based on a combination of existing state-of-the-art techniques, but the specific combination approach is non-trivial and highly interesting.
The experimental setup is also practical, making it well-prepared for discussion from diverse perspectives within the community.
For these reasons, I believe this paper is worthy of presentation within the relevant community.

---

> ### Author Rebuttal · Authors · 2025-10-18
>
> Thank you for your thoughtful and constructive feedback.
>
> Regarding your first comment, could you elaborate on which part (handling variable lengths) specifically gives that impression?
>
> Regarding the second comment, our aim in this paper was to investigate the proposed architecture, assess its effectiveness, and demonstrate a proof-of-concept on a specific class of instruments. We chose bass and guitar because they are fundamental melodic and harmonic components in many contemporary genres, and they exhibit distinct sonic characteristics—bass in the low-frequency spectrum and guitar in the mid-to-high frequencies. As a result, by isolating variables, we were able to better understand how our model learns representations for instruments with different timbral and frequency profiles. Furthermore, training on a more focused dataset (e.g., only bass loops) provides a clearer definition of "normalcy," making the task of identifying meaningful anomalies more tractable for this initial investigation. This also allowed us to create a more stable and interpretable environment for tuning the Deep SVDD module.
>
> It is important to note that, while we focused on these instruments for the study, the architecture itself is instrument-agnostic. The pre-trained HTSAT model was trained on a wide variety of sounds, including music and diverse audio events, and is not specialized for string instruments. We are therefore confident that the proposed pipeline can be generalized to other materials.
>
> Moreover, we agree with the reviewer that demonstrating the system's performance on more diverse sources like vocals or drums would be a compelling extension of this work. We have noted this as a direction for future work in Section 9 of the paper.

---

### Official Review · Reviewer_BdVZ · 2025-10-10
**An unsupervised method for anomaly detection in musical patterns**

**Rating:** 5
**Confidence:** 3
**Final Rating:** 4
**Final Confidence:** 4

**Summary:**

The paper presents an unsupervised method for detecting anomalous patterns in musical loops. The method can deal with variable input via feature fusion mechanism (ie. padding if sequence shorter, and cutting and subsampling followed by joining with attention mechanism if sequence is longer). The inputs are encoded with a pretrained Hierarchical Token-semantic Audio Transformer (HTS-AT) encoder, and embeddings analyzed by Deep Support Vector Data Description (unsupervised encoding into a hypersphere, where anomalous points are outside). The authors report a better anomaly detection compared to baseline.
The anomaly scores are evaluated against IF and PCA, since no ground truth is available.

**Strengths:**

The paper is well structured and very well written, easy to follow. The method presented is a collection of modules that are in themselves not new but well put together, and it tackles unsupervised anomaly detection, which has been underexplored in the musical pattern recognition.

**Weaknesses:**

There are only a few minor comments/questions that arised:

Too bad that data is not available, for reproducibility. It would be interesting to rerun the analysis on some open dataset also, and perhaps even  with partial labels.

You only compared your pipeline against PCA and IF. It would be interesting to see a comparison with another DL model (eg form 14), perhaps combined with FFM, to see if it's the Deep SVDD that plays the most important role here...

How is the "better anomaly detection" evaluated? - I would like to have a sentence on that already where you describe the different plots/result you are going to show.

You write that for sequences longer than d seconds, three d-second segments are randomly sliced from the beginning (first 1/3), middle (second 1/3), and end (final 1/3) of the clip. But just because a sequence is longer than d it doesn't mean that cutting it to thirds will produce three segments of length d. Please explain how exactly they are created in your case (eg. Do you cut by thirds and then pad/repeat, like with those shorter than d? What if the segment is much longer than 3d? ...). Also why use thirds in specific and, since FFM since to be quite useful in the piplein, could it potentially help if you used multiple scales for downsampling?

 In 4.3. you write "In AE pre-training, the autoencoder learned a compressed data representation by training on pre-computed embeddings with a Mean Squared Error (MSE) reconstruction loss." Can you elaborate on this - how were the embedding precomputed then?  Is the autoencoder not trained first in a usual way, learning to reconstruct the images themselves?

When you explain how the methods are evaluated you mention a number of plots that are not included in the paper. Make it clear where the plots are (that they are in the appendix for example).

In the results you write that "AEwRES showing the most compact distribution and strongest outlier separation." Can you elaborate - to me it seems like the normal AE seems more compact (based on the plots).

**Final Justification:**

The evaluation could be more elaborate, both in terms of more data and data with potential ground truth, to make it more quantitative and comparable. The paper would benefit from some minor corrections.
However, the authors have adequately answered the comments and the paper presents a pipeline that is new and deals with an underexplored problem, the writing is clear and easy to follow. A new and interesting  development worth sharing.

**Justification:**

The paper is well written and presents an interesting pipeline as well as evaluation approach.
It tackles unsupervised anomaly detection in musical segments, which has been underexplored.

---

> ### Author Rebuttal · Authors · 2025-10-18
>
> Thank you for your thoughtful and specific questions.
>
> Regarding reproducibility, we plan to release the source code.
>
> For the comparison, the goal was to show the value of the integrated framework versus simpler methods, not to find the absolute best anomaly detection algorithm. IF was used to test whether a specialized deep learner offers an advantage over a standard ensemble method, while PCA-based reconstruction error checked if non-linear models outperformed linear ones. By using the same HTS-AT embeddings for all methods, the benefit of the Deep SVDD objective could be clearly observed.
>
> Regarding your third comment, you are correct that an upfront definition of how “better” is measured would improve the paper’s clarity. To elaborate, in this study, “better anomaly detection” is not evaluated with a single quantitative score (due to lack of labels) but is a holistic assessment based on the qualitative characteristics of the learned latent space. In that regard, a model is considered “better” if it demonstrates (1) a clear latent space separation (e.g., a tighter, more cohesive cluster of “normal” samples), (2) a more informative representation (e.g., a higher explained variance ratio), and (3) higher anomaly score distribution (e.g., clear margin between normal and anomalous data). We agree that explicitly stating these criteria at the beginning of the "Evaluation" Section would have been beneficial for the reader.
>
> Thank you for this question (fourth comment). It raises a subtle implementation detail. We recognize that the paper's description could be improved for clarity. The core idea is to get representative samples from the beginning, middle, and end of the loop to capture its full temporal character [19]. Below, I will explain this in more detail:
>
> - To begin, we first address *how the segments are created*. For an audio clip of total length L>d, the clip is conceptually divided into thirds. A *d*-second segment is then randomly sampled from within each of these three sections. For instance, a *d*-second window is chosen from the time interval [0, L/3], another from [L/3, 2L/3], and a third from [2L/3, L]. This guarantees each segment is exactly d seconds long and that the start, middle, and end are all represented.
>
> - If the clip is much longer than 3d, we still sample randomly from each third. This is a computationally efficient way to get representative snapshots without processing all the audio at high resolution.
>
> - Your intuition is correct about multi-scale downsampling. The pipeline uses three high-resolution local segments and one globally downsampled d-second version of the clip. The attention feature fusion part learns to weigh global context against local details.
>
> Regarding your fifth comment, you are highlighting the sequential, two-stage pipeline. In stage 1, we extract features. All audio samples are processed through a feature fusion mechanism and then passed to the pretrained and frozen HTSAT model. This outputs a 1024-dimensional embedding for each audio input before training begins. These are the "pre-computed embeddings." In stage 2, we train the anomaly model. The autoencoder (used for Deep SVDD) trains only on the 1024-dimensional embeddings. Its encoder further compresses the rich HTSAT features into a compact latent space that captures "normal" loops. The autoencoder does not reconstruct audio; it learns compressed representations of HTS-AT model features.
>
> Thank you for pointing this out (sixth comment). We will clarify the location of the plots.
>
> Thank you for highlighting this point (last comment). This requires a deeper interpretation of the results. In short, the AE model produces anomaly scores in a smaller absolute range, which makes its distribution *appear* more compact. On the other hand, the AEWRES model demonstrates better feature separability, which is more desirable for effective anomaly detection. To elaborate, the AE model’s compactness on the guitar dataset is likely a sign of over-compression. In other words, it learns a simplistic representation that collapses distinct but normal patterns together; it loses nuance. This is also supported by its lower EVR (77.7%) compared to AEWRES EVR (93.2%). Furthermore, looking at PCA plots (e.g., Figures F1 and F2), the AEWRES model achieves a cleaner and more distant separation between the main cluster of normal samples (blue) and the identified anomalies (red).

---

### Official Review · Reviewer_JZWr · 2025-10-13
**Not enough musically grounded unsupervised anomaly detection pipeline for analyzing variable-length musical loops**

**Rating:** 2
**Confidence:** 3

**Summary:**

Authors present their unsupervised anomaly detection framework for analyzing musical loops specifically addressing challenges of variable-length music pieces and the lack of labeled data. Proposed pipelines consists of pre-trained HTS-AT transformer for feature extraction, Feature Fusion Mechanism to handle different input lengths, and Deep SVDD for identifying anomalous patterns. Authors run experiments using their pipeline on guitar and bass loops to show that proposed method outperforms traditional methods like Isolation Forest and PCA.

**Strengths:**

This paper is very interesting, sound, and exploratory and it fits in the field of music modelling and potentially music mining. Objectives and goals are both underexplored and nobel. The paper is very detailed and provides most of information for reproducibility. Authors spend a good amount of space for discussion and limitations of their work

**Weaknesses:**

I am not a musician myself, but I know that music is subjective and is created for various purposes. Authors claim that the system that can flag loops with clicks, phase issues, noise. Artifacts in music are context-dependent and might be intentional. Therefore, proposed pipeline's effectiveness would depend on diversity and quality of training data. Authors do not mention how false-positives are processed or analysed to justify model's accuracy. I think that the paper is overstating readiness of the method. Evaluation relies on PCA visualisation and latent representations without ground truth labeling. Qualitative methods are ok for unsupervised methods for initial exploration. However, more rigor is needed for objective evaluation. Are anomalies detected by the pipeline musically meaningful or just statistical outliers? Another issue I have with using thresholding is that it assumes that certain % (5%) of the data is anomalous without having any musically-grounded justification

**Justification:**

The main guidance that I have for my decision is the lack of objective evaluation and musically-grounded analysis of results. Otherwise the paper is interesting and has potential

---

> ### Author Rebuttal · Authors · 2025-10-18
>
> Thank you for your thoughtful and constructive feedback. This issue is a core challenge in unsupervised learning, especially in creative domains like music. We would like to take this opportunity to clarify the intended purpose of our system and the rationale behind our evaluation methodology, which we believe addresses the reviewer's valid concerns.
>
> We acknowledge that musical artifacts can be intentional, and distinguish between our framework's assistive role and automated decision-making. In Section 7.1, we position it for flagging potential issues for human review, not automatic rejection. The unsupervised nature allows adaptation to different musical contexts. For instance, a model trained on lo-fi recordings would learn different "normal" patterns than one trained on polished productions. This setup naturally accommodates genre-specific intentional artifacts.
>
> The reviewer correctly notes that the system's effectiveness depends on the training data. Indeed, this is an intended feature that allows the system to adapt to a user's personal sample library or a specific project's stylistic nuances. By learning the "normal" patterns of a given collection, it can effectively highlight deviations specific to that context.
>
> Regarding subjectivity, the pipeline’s role is to flag what is *different*. It leaves the judgment of whether that difference is “good” (creative and unique) or “bad” (a technical flaw) to the user. In this regard, the paper suggests applications in both quality control (finding errors) and creative discovery (finding unique sounds). In this context, a “false positive” for quality control could be a “true positive” for creative discovery. For instance, a loop flagged for an unusual harmonic texture might be exactly what a practitioner is looking for to add a unique element to their track. In short, the system automates the search for the unusual, but the user provides the final, subjective interpretation.
>
> We acknowledge the limitations of an evaluation without ground-truth labels. However, our methodology provides strong evidence in the pipeline's capacity to detect musically meaningful irregularities. This is supported by both its architectural design and the structured, interpretable nature of its outputs. The components of the proposed architecture are chosen to capture and extract the audio’s inherent structure. The system operates on musical features, not raw statistics. The HTSAT model, pre-trained on massive audio datasets including music, is central to this. Together with the feature fusion mechanism (FFM), they extract features (embeddings) that carry a semantically rich representation of musical characteristics.
>
> Regarding the Deep SVDD (anomaly detector), its specific objective is to learn a transformation mapping the “normal” data into a compact hypersphere within a latent space. This approach distinguishes between general statistical outliers in the initial feature space and musical anomalies whose structure deviates from learned norms. Here, an “anomaly” refers not simply to a statistical outlier, but to a musical loop with structural differences from the established norm.
>
> The latent space visualization plots provide evidence for this ability to learn a musically relevant manifold. The PCA plots (e.g., Figures F.1 and F.2) show that the AEWRES model, in particular, does not create a random cloud of “normal” points. On the guitar dataset, it learns a distinct, curved structure for the normal samples. This suggests that the model has captured the underlying “rules” or continuous variations of a *typical* guitar loop in the *dataset*. The high explained variance ratio (EVR) captured by the first few principal components (e.g., 93.2% for AEWRES on the guitar dataset) also indicates that the learned structure is not arbitrary but instead captures the dominant mode of variation in data.
>
> We appreciate the reviewer’s skepticism of the fixed 95% percentile threshold. The threshold value serves as a heuristic separation point rather than a verdict that 5% of the data is flawed. Indeed, its primary function is to provide a manageable starting point for exploration by ranking the loops from typical to most unusual. The system's core output is not a binary classification but a *continuous* anomaly score for each loop. This score represents its distance from the learned center of “normalcy”. It enables a flexible workflow where a user can rank their collection by this score to explore the sample collection. They are not constrained by an arbitrary cutoff. Indeed, the threshold was used in our experiments to provide a consistent basis for comparing distributions of the most anomalous samples generated by different models (AE, AEWRES, IF, PCA reconstruction). With this approach, the system empowers the user to apply their own judgment to a ranked list of possibilities, rather than requiring a specific, “musically-grounded” threshold justification.
>
> Moreover, we acknowledge that qualitative evaluation with musicians (perceptual evaluation) would strengthen our claims further. Nevertheless, we view this work as establishing the foundation and demonstrating a proof-of-concept. Therefore, the perceptual evaluation with human subjects is a separate, application-focused research question that represents a logical next phase of research.
>
> In essence, our confidence stems from the pipeline's ability to learn and represent the shared musical characteristics of an audio collection, thereby distinguishing samples based on their deviation from this learned essence. The threshold is a practical tool that facilitates human exploration of these deviations, rather than a rigid classifier. This distinction underscores the pipeline’s role in assisting, not dictating, human creative decisions.

---

### Meta-Review · Area_Chair_Gfm1 · 2025-10-31

**Recommendation:** Accept (Poster)
**Confidence:** 3

**Metareview:**

This paper presents a unsupervised method for the analysis of audio loops, combining feature extraction (FFM + HTS-AT) and anomaly detection techniques (DeepSVDD) in order to be able to classify "normal" vs. "anomalous" loops. The framework introduced in the paper does not rely on handcrafted features, and is not limited by variable length inputs. The authors report an improvement in anomaly detection compared to baselines IF and PCA.

The reviewers agreed that the presented architecture, combining existing methods from different fields, is clearly described and provides an interesting solution to the problem under consideration. The method used for features extraction allows for variable-length inputs, which is often a limitation for many methods. The experimental setup is described in details, and authors plan to release the source code of their experiments, which will help reproducing the experiments.
The reviewers however pointed that despite the promising results, the paper providing little insight about the detected anomalies. The dataset used in the paper, in addition to not being publicly available, is very restricted (bass and guitar only). This was acknowledged by the authors, which described the work as a "proof of concept".
The authors provided the expected clarifications during the rebuttal phase, and given the reviewer's impressions the paper can be accepted.

---

### Decision · Program_Chairs · 2025-11-05

**Decision:**

Accept (Spotlight)

**Comment:**

We recommend an oral and a poster presentation given the AC and reviewers recommendations.

A spotlight presentation refers to a poster selected for an oral highlight but not designated as a full oral presentation per the AC’s recommendation.